# Large-Scale Transcriptome Profiling and Network Pharmacology Analysis Reveal the Multi-Target Inhibitory Mechanism of Modified Guizhi Fuling Decoction in Prostate Cancer Cells

**DOI:** 10.3390/ph18091275

**Published:** 2025-08-27

**Authors:** Guochen Zhang, Lei Xiang, Qingzhou Li, Mingming Wei, Xiankuo Yu, Yan Luo, Jianping Chen, Xilinqiqige Bao, Dong Wang, Shiyi Zhou

**Affiliations:** 1School of Basic Medical Sciences, Chengdu University of Traditional Chinese Medicine, Chengdu 611137, China; zhangguochen@stu.cdutcm.edu.cn (G.Z.); leixiang@stu.cdutcm.edu.cn (L.X.); liqingzhou@stu.cdutcm.edu.cn (Q.L.); weimingming688@163.com (M.W.); 2020kb028@stu.cdutcm.edu.cn (X.Y.); 2020ks053@stu.cdutcm.edu.cn (Y.L.); 2LKS Faculty of Medicine, School of Chinese Medical, The University of Hong Kong, Hong Kong 999077, China; abchen@hku.hk; 3Medical Innovation Center for Nationalities, Inner Mongolia Medical University, Hohhot City 010110, China; 20200005@immu.edu.cn

**Keywords:** prostate cancer, Modified Guizhi Fuling Decoction, HTS^2^ technology, network pharmacology, transcriptomics, bioinformatics

## Abstract

**Background:** Prostate cancer (PCa) is the primary contributor to male cancer-related mortality and currently lacks effective treatment options. The Modified Guizhi Fuling Decoction (MGFD) is used in clinical practice to treat multiple tumors. This research focused on the mechanisms of action (MOA) in MGFD that inhibit PCa. **Methods:** The impact of MGFD on PCa cells (PC3 and DU145) was examined via Cell Counting Kit-8, wound healing assays, and transwell assays. To determine the MOA, high-throughput sequencing based high-throughput screening (HTS^2^) was utilized along with network pharmacology. **Results:** The findings indicated that MGFD suppressed the proliferation, migration, and invasion of PCa cells. We then utilized the HTS^2^ assay to generate 270 gene expression profiles from PCa cells perturbed by MGFD. Large-scale transcriptional analysis highlighted three pathways closely associated with PCa: the TNF signaling pathway, cellular senescence, and FoxO signaling pathway. Through the combination of network pharmacology and bioinformatics, we discovered four primary targets through which MGFD acts on PCa: AKT serine/threonine kinase 1 (AKT1), Caspase-8 (CASP8), Cyclin-Dependent Kinase 1 (CDK1), and Cyclin D1 (CCND1). Finally, molecular docking demonstrated that the potential bioactive compounds baicalein, quercetin, and 5-[[5-(4-methoxyphenyl)-2-furyl] methylene] barbituric acid strongly bind to CDK1, AKT1, and CASP8, respectively. **Conclusions:** This research shows that MGFD displays encouraging anticancer effects via various mechanisms. Its multi-target activity profile underscores its promise as a potential therapeutic option for PCa treatment and encourages additional in vivo validation studies.

## 1. Introduction

Prostate cancer (PCa) ranks as the second most prevalent cancer, trailing only behind lung cancer in men [1,2]. Statistics highlight the significant public health challenge posed by PCa, which is responsible for more than 300,000 fatalities annually, making it one of the leading causes of cancer-associated fatalities in the male population [3]. The impact of PCa on men's health is profound, highlighting the urgency for prevention, early detection, and treatment. Furthermore, PCa is characterized as a hormone-dependent malignancy, which means that its growth and progression are closely linked to hormonal influences, particularly androgens. Androgen signaling is essential not only for normal prostate development but also for the PCa advancement [4]. Androgen deprivation therapy (ADT) serves as the main approach for treating PCa. However, as the disease progresses, an increasing number of patients develop castration-resistant prostate cancer (CRPC), in which tumors continue to grow and metastasize despite ADT, significantly increasing clinical mortality [5]. Currently, effective therapeutic options for CRPC remain limited, and the molecular mechanisms underlying treatment resistance are not fully understood. Therefore, identifying novel drugs and therapeutic targets for PCa, has become a critical focus of ongoing research.

Traditional Chinese medicine (TCM) boasts a rich and extensive history in cancer treatment, which has garnered significant interest in recent years. This interest is particularly evident in the exploration of how TCM can be integrated with modern medical practices [6,7]. This approach offers a unique perspective on cancer care, potentially enhancing treatment outcomes and providing patients with a more holistic approach to their health. As researchers and practitioners continue to investigate the synergies between TCM and modern medicine, the potential benefits of such integrative strategies are becoming increasingly recognized within the medical community. The Modified Guizhi Fuling Decoction (MGFD) incorporates additional herbs such as *Cullen corylifolium* (L.) Medik. (Buguzhi) and *Drynaria roosii* Nakaike (Gusuibu) into the traditional Guizhi Fuling Decoction (GFD), a formula originating from Zhang Zhongjing's Jin Gui Yao Lue. The GFD formula includes *Neolitsea cassia* (L.) Kosterm. (Guizhi), *Poria cocos* (Schw.) Wolf. (Fuling), *Paeonia lactiflora* Pall. (Chishao), *Paeonia × suffruticosa* Andrews (Mudanpi), and *Prunus persica* (L.) Batsch (Taoren), which promote blood circulation, resolve stasis, and dissipate masses [8,9]. Furthermore, Buguzhi and Gusuibu are frequently prescribed to alleviate bone metastatic cancer pain [10]. The extract from Buguzhi has demonstrated effectiveness in diminishing bone metastasis within models of breast cancer [11]. In clinical settings, Chinese doctors frequently employ MGFD to address a range of tumors, including those of the endometrium, ovaries, and prostate cancer. Nevertheless, the molecular mechanisms through which MGFD operates in the treatment of PCa have not been thoroughly investigated.

High-throughput sequencing based high-throughput screening (HTS^2^), which combines the RNA annealing, selection, and ligation (RASL) approach with next-generation sequencing technology. This method can simultaneously detect thousands of gene expressions in cells under various perturbations [12,13]. HTS^2^ technology is particularly effective for the pathway-centric discovery of anticancer drugs [14]. It also can enhance our understanding of herbal treatments by providing extensive gene expression profiles. When paired with network pharmacology, it elucidates the intricate molecular interactions involved in herbal anticancer therapies [15,16]. Additionally, network pharmacology, a multidisciplinary approach that merges bioinformatics with other analytical methods, has greatly advanced TCM research by elucidating complex mechanisms and identifying bioactive compounds in herbal formulations [17,18].

This study aims to systematically investigate the inhibitory effects of MGFD on human PCa cells and elucidate the underlying molecular mechanisms. Initially, to evaluate the regulatory effects of MGFD on PCa cells, related cellular experiments were conducted. Subsequently, HTS^2^ technology was employed to obtain large-scale gene expression profiles of human PCa cells treated with MGFD. Concurrently, key pathways and targets of MGFD in PCa were identified using bioinformatics and network pharmacology approaches. Finally, molecular docking was used to validate the interactions between potential bioactive compounds and the identified targets.

## 2. Results

### 2.1. MGFD Inhibits the Proliferation, Migration, and Invasion of PCa Cells

The cell viability of MGFD-extract-treated DU145 and PC3 cells was measured. The 50% inhibitory concentration value for DU145 and PC3 cells were 118.2 μg/mL and 104.1 μg/mL, respectively (Figure 1A). These results indicated a reduction in the cellular activity of PCa cells with increasing concentrations of MGFD. Following that, the inhibitory impact of MGFD on the migration of DU145 and PC3 cells at two concentrations (high [75 μg/mL] and low [50 μg/mL]) was confirmed (Figure 1B). The invasion and migration of these cells were effectively suppressed by MGFD through the transwell invasion and migration assays. This inhibitory effect was observed at two concentrations (high and low) (Figure 1C).

### 2.2. Construction of 270 Gene Expression Profiles from MGFD-Perturbed Prostate Cancer Cells Through HTS^2^ Assay

Through the application of the HTS^2^ assay, we produced extensive gene expression data encompassing 3407 genes associated with cancer. This includes 270 distinct gene expression profiles and a total of 919,890 gene expression events influenced by MGFD. The comprehensive findings can be found in Appendix A. The results obtained from Pearson's correlation test indicated that the scores for both the positive control and the blank groups exceeded 0.9, demonstrating the high reliability and technical stability of the HTS^2^ assay (Appendix A). Meanwhile, some of the sequencing results were validated by quantitative real-time polymerase chain reaction (qRT-PCR) experiments, demonstrating the accuracy of the sequencing data (Appendix A). Furthermore, a heatmap illustrated the gene expression profiles after MGFD perturbation in DU145 and PC3 cells (Figure 2A). By analyzing the differentially expressed genes (DEGs) in the drug-treated groups at each concentration, we observed that the number of DEGs increased with higher drug concentration, suggesting a potentially enhanced anti-tumor effect (Figure 2B, Appendix A). Consequently, we selected the treatment group with a drug concentration of 100 μg/mL for further analysis. Appendix A shows the cell viability of the seven herbal extracts at a concentration of 100 μg/mL in both cell lines.

### 2.3. Buguzhi and Guizhi Show More DEGs than Those of Other Herbs in MGFD

Buguzhi and Guizhi exhibited a greater number of DEGs than other herbs in MGFD (100 μg/mL). The expression profiles of genes revealed that the DU145 cell line showed the greatest quantity of DEGs for Buguzhi, comprising 161 upregulated and 117 downregulated genes. Chishao had the lowest number of DEGs, with 33 genes, including 16 upregulated and 17 downregulated genes. Other herbs showed varying numbers of DEGs, including Guizhi (165), Gusuibu (69), Mudanpi (141), Fuling (74), and Taoren (54) (Figure 3A, Appendix A). Similarly, in the PC3 cell line, the highest number of DEGs for Buguzhi was observed, with 216 upregulated and 205 downregulated genes (Figure 3B). Taoren had the lowest number of DEGs, with a total of 37, including 22 upregulated and 15 downregulated. Other herbs exhibited the following numbers of DEGs: Guizhi (185), Fuling (63), Mudanpi (77), Gusuibu (56), and Chishao (42). A comparison of the DEGs between the two cell lines revealed that Buguzhi and Guizhi had a greater number of DEGs. The UpSet plot visualizes the overlap and specificity of DEGs between the two cell lines (Appendix A).

Cluster analysis of the DEGs was conducted to explore the relevance of each herb in the MGFD. The results showed that the seven herbs of MGFD could be categorized into three main groups: Buguzhi and Guizhi in group 1; Gusuibu and Fuling in group 2; and Mudanpi, Chishao, and Taoren in group 3. (Figure 3C). According to TCM theory, the Jun-Chen-Zuo-Shi principle highlights the synergistic effects of herbal formula to improve treatment efficacy. In this formula, Buguzhi and Guizhi act as sovereign herbs providing the main therapeutic effect. Gusuibu and Fuling serve as minister herbs, enhancing the action of the sovereign herbs and balancing the formula. Mudanpi, Chishao, and Taoren function as assistant and courier herbs, harmonizing the overall effect and supporting the treatment.

### 2.4. Investigation of Effect of Each Herb in MGFD on Gene Sets

The impact of each herb in MGFD on tumors was analyzed using a Sankey network, integrating the pathway-related biological functions revealed by Gene Set Enrichment Analysis (GSEA) based on hallmark gene sets (Figure 4A,B, Appendix A). The gene sets enriched by the seven herbal medicines primarily demonstrated significant anti-tumor properties through several mechanisms. The key effects include the inhibition of cell proliferation, migration, and invasion, along with apoptosis promotion. Furthermore, the gene set from these herbs is involved in hindering the cell cycle. Furthermore, Buguzhi and Guizhi enriched more gene sets in both cell lines compared to other herbs, including “Apoptosis”, “CELL CYCLE”, and “MYC target V1”, suggesting primary role of Buguzhi and Guizhi in MGFD.

UpSet plots were used to visually present the enriched gene sets of the seven herbs across different cell lines, facilitating comparison of their similarities and differences (Appendix A). In the DU145 cell line, Buguzhi, Guizhi, Chishao, Mudanpi, Gusuibu, and Fuling all upregulated the “P53 pathway” gene set, which is closely associated with apoptosis of tumor cells (Figure 4C). Additionally, Buguzhi, Guizhi, Mudanpi, and Fuling downregulated the “MYC targets v1” and “G2M checkpoint” gene sets, which are closely related to the inhibition of tumor cell proliferation and cell cycle progression. Meanwhile, in the PC3 cell line, Buguzhi, Guizhi, Chishao, and Mudanpi upregulated the “Cholesterol Homeostasis” gene set, thereby modulating dysregulated lipid metabolism in tumor cells (Figure 4D). Furthermore, Buguzhi, Guizhi, and Taoren downregulated the “E2F targets” and “G2M checkpoint” gene sets, which are closely involved in cell cycle arrest.

### 2.5. The Kyoto Encyclopedia of Genes and Genomes (KEGG) Pathway Network Enrichment Analysis Identified Three Hub Pathways

KEGG enrichment analysis was performed for each of the seven herbs in MGFD, and the Herb-KEGG interaction network was established. Topology analysis revealed the fifteen pathways exhibiting the highest degree scores across each cell line, as illustrated in Figure 5A,B, and detailed in Appendix A. Notably, eight pathways were highly ranked in both PC3 and DU145 cell lines (Figure 5C). Three pathways closely related to tumor development were selected from these eight pathways as hub pathways for MGFD in the treatment of PCa: TNF signaling pathway (hsa04668), cellular senescence (hsa04218), and FoxO signaling pathway (hsa04068). Genes associated with key pathways were utilized to build a PPI network, and the 20 genes exhibiting the highest degree scores were recognized as central targets (Figure 5D, Appendix A).

The biological functions of the DEGs in the MGFD were further elucidated through Gene Ontology (GO) analysis. The DEGs in both cell lines were primarily involved in the cellular response to chemical stress/external stimuli, apoptotic signaling pathway regulation, cell cycle, and wound healing (Figure 5E,F, Appendix A).

### 2.6. Active Component-Target Network Construction and Its Enrichment Analysis Through Network Pharmacology

Initially, 79 active components associated with the MGFD were extracted from the databases. These components were selected based on screening conditions such as OB ≥ 30% and DL ≥ 0.18. Subsequently, SwissTargetPrediction identified the drug targets associated with these active components, resulting in 691 drug targets after eliminating redundant targets (Appendix A). Disease-specific targets of PCa were identified using databases such as DisGeNET, OMIM, and Genecards (Appendix A). A total of 418 disease targets for PCa were identified by summarizing the targets mapped between each pair of databases (Figure 6A). The mapping of disease targets to drug targets led to the identification of 84 therapeutic targets (Figure 6B).

To illustrate the relationship between active components and potential therapeutic targets, an active-target network for MGFD was created with the help of Cytoscape. This network comprised 84 PCa-associated targets and 79 active components (Figure 7A). KEGG enrichment analysis results showed that MGFD acts on PCa by affecting pathways such as “pathways in cancer”, “prostate cancer”, and “endocrine resistance” (Figure 7B). Furthermore, the analysis of GO revealed that the 84 targets associated with PCa played a significant role in biological processes, including positive and negative regulation of “signal transduction”, “apoptotic process”, and “RNA polymerase II promoter transcription” (Figure 7C).

### 2.7. Genetic Alterations and Survival Analysis of These Therapeutic Targets in PCa Patients

To delve deeper into the molecular signatures associated with the 84 therapeutic targets among various cohorts of PCa patients, analyses of genetic alterations and survival were conducted utilizing cBioPortal. The findings revealed that genetic alterations were observed in roughly 50% to 90% of PCa patients across the six different datasets. Gene alterations are more frequent in metastatic PCa than in nonmetastatic PCa. Notably, 84 therapeutic targets primarily exhibited mutations in non-metastatic PCa, whereas in metastatic PCa, they mainly showed multiple alterations and amplifications (Figure 8A). Furthermore, patients with genetic alterations exhibited lower survival rates than those without these alterations (Figure 8B). The findings indicate a significant relationship between the 84 therapeutic targets and the prognosis of PCa, thus offering additional backing for the use of MGFD in the treatment of this condition.

### 2.8. Disease-Free Survival Analysis of These Four Key Targets in PCa Patients

The gene expression profiles from the HTS^2^ assay identified 20 hub targets, which were mapped to 84 therapeutic targets obtained from network pharmacology analysis, resulting in four key targets (AKT1, CASP8, CDK1, and CCND1) for the MGFD treatment of PCa (Figure 9A). Subsequently, the analysis of disease-free survival revealed that, with the exception of CCND1, the *p*-values for AKT1, CASP8, and CDK1 were all below 0.1. Furthermore, elevated expression levels of these genes were significantly linked to unfavorable outcomes for patients (Figure 9B).

### 2.9. Disclosing Bioactive Components of MGFD Against Key Targets

To validate previous research findings, molecular docking was employed to evaluate the selected bioactive components and their corresponding key targets. The binding energies obtained from the docking of these bioactive compounds and key targets are presented in Table 1, where these energies act as indicators of the docking capability. Compounds with Glide gscore below −7.0 were considered to have good binding potential. Baicalein, quercetin, and 5-[[5-(4-methoxyphenyl)-2-furyl] methylene] barbituric acid exhibited the highest docking activities with CDK1, AKT1, and CASP8, respectively. Diagrams depicting three-dimensional interactions were employed to demonstrate the interactions between the active components and their targets (Figure 10A–C). However, because of the absence of direct binding inhibitors of CCND1, it was excluded from the molecular docking process.

## 3. Discussion

The global healthcare system is facing a significant challenge in effectively treating PCa owing to its rising incidence associated with an aging population. Despite the approval of various novel drugs in recent years, the mortality rate associated with PCa remains high [25]. Consequently, we primarily focused on the development of treatments for PCa. Recent studies have demonstrated that GFD can inhibit cervical cancer invasion by restoring the balance of MMP-TIMPs and suppress breast cancer by targeting the PI3K and MAPK pathways [26,27]. Moreover, Psoralen induces cell death through endoplasmic reticulum stress by enhancing pancreatic endoplasmic reticulum kinase activity [11]. Rhizoma Drynariae can enhance tumor immune function by up-regulating IL-2 levels [28]. Drawing from these discoveries, MGFD has promising applications in the clinical treatment of PCa.

In this research, we performed phenotypic experiments to show that MGFD hinders the growth and invasion of DU145 and PC3 cells. Notably, the large-scale gene expression profiles generated through the HTS^2^ assay helped elucidate how MGFD functions to inhibit PCa cells through various pathways and targets. A concentration of 100 μg/mL for seven herbs typically resulted in a greater quantity of DEGs, implying that this level might be ideal for achieving therapeutic benefits. At this concentration, we conducted GSEA and heatmap cluster analysis. The results showed that in the MGFD, Guizhi and Buguzhi were the primary anti-tumor herbs that functioned as sovereign herbs. The majority of the gene sets that are regulated were linked to the prevention of proliferation and invasion, the promotion of apoptosis, and the halting of the cell cycle. The remaining five herbs in the decoction contributed to the anti-PCa effects from various perspectives, including classical anticancer, immune regulation, and lipid metabolism, serving as minister and assistant herbs.

MGFD inhibited the growth of PCa cells mainly through three pathways: FoxO signaling pathway, cellular senescence, and TNF signaling pathway. The FoxO signaling pathway is essential for numerous cellular function regulations, including apoptosis, control of the cell cycle, and responses to oxidative stress [29]. Disruption of this pathway is linked to the onset and advancement of multiple cancers, including PCa [30,31]. Cellular senescence refers to a condition characterized by a permanent halt in the cell cycle, which occurs when cells react to different stress factors, including DNA damage or the activation of oncogenes [32]. This phenomenon serves as a powerful mechanism for tumor suppression by inhibiting the growth of compromised cells. Nevertheless, senescent cells have the potential to facilitate tumor advancement via a senescence-associated secretory phenotype (SASP), thereby rendering the targeting of these senescent cancer cells a particularly appealing strategy for therapy [33,34]. The signaling pathway of TNF is significant in both inflammation and immune responses [35]. Research indicates that TNF-α is crucial for processes of tumor proliferation, migration, invasion, and angiogenesis [36]. In cancer, TNF signaling may exhibit dual functions, either facilitating the death of tumor cells or supporting their survival, which is contingent upon the cellular environment and the interplay of subsequent signaling events [37].

By combining transcriptional analysis with network pharmacology, we identified four key targets: AKT1, CASP8, CDK1, and CCND1. AKT1 is a serine/threonine kinase that phosphorylates specific residues of FOXO transcription factors. This mechanism causes the movement of FOXO from the nucleus into the cytoplasm, which in turn reduces its ability to activate transcription. As a result, cancer cell survival and invasion are promoted by downregulating the genes responsible for FOXO-induced apoptosis and cell cycle arrest [38,39]. Targeting AKT1 or restoring FOXO function is considered a potential therapeutic strategy due to the significant function of the AKT1-FOXO relationship in cancer progression [40]. CASP8, a cysteine protease of the caspase family, is significant in apoptosis, particularly through the extrinsic pathway mediated by death receptors, and its abnormal expression has been linked to poor prognosis in PCa patients [41,42]. CASP8 is a potential target for counteracting enzalutamide resistance in CRPC [43]. Additionally, CDK1 and CCND1, which are key players in cell cycle control in PCa, regulate the G2/M and G1/S phases, respectively [44]. Furthermore, CDK1 and CCND1 are influenced by the FoxO signaling pathway, in which FOXO nuclear translocation inhibits the cell cycle, leading to reduced PCa cell proliferation [45]. In conclusion, all four key targets were closely related to the FoxO signaling pathway, highlighting its significant role among the hub pathways.

The findings from molecular docking suggested that the primary bioactive compounds of MGFD, which may have the ability to inhibit PCa cells, encompass baicalein, quercetin, and 5-[[5-(4-methoxyphenyl)-2-furyl] methylene] barbituric acid. Baicalein and quercetin are members of the flavonoid family and demonstrate a diverse array of biological functions [46]. Quercetin has been reported to inhibit the migration of human peritoneal melanoma cells by suppressing AKT phosphorylation [47]. In vivo studies further demonstrate that quercetin induces apoptosis in hepatocellular carcinoma cells through inhibition of the PI3K/Akt/mTOR signaling pathway [48]. Moreover, baicalein and its derivatives have been shown to inhibit the CDK1 protein, with certain derivatives—such as 8-hydroxypiperidinemethyl-baicalein—identified as potent CDK1 inhibitors [49]. However, no experimental studies are currently available for 5-[[5-(4-methoxyphenyl)-2-furyl] methylene] barbituric acid, and its biological activity remains to be further explored.

TCM is characterized by multiple pathways, targets, and components for disease treatment, leading to synergistic effects [50,51]. By analyzing large-scale gene expression profiles using HTS^2^ technology, researchers can identify hub pathways and targets of TCM in disease treatment. Meanwhile, network pharmacology can reveal the potentially bioactive components of TCM. Collectively, this research deepens our comprehension of TCM theories, clarifies the intricate molecular processes associated with medicinal herbs, and uncovers possible bioactive compounds.

However, several limitations remain in this study. First, the lack of in vivo experimental validation restricts a comprehensive assessment of MGFD’s antitumor efficacy within the complex physiological environment. Second, this study primarily relied on the HTS^2^ technology for mechanistic exploration. Although this approach provides extensive transcriptomic data, its depth in elucidating molecular mechanisms is still limited. Future studies should incorporate additional experimental methods, such as proteomics and functional assays, to strengthen the robustness of the conclusions. Furthermore, the pharmacokinetic and pharmacodynamic properties of the active components in MGFD have not been systematically investigated, and knowledge regarding their absorption, distribution, metabolism, and excretion (ADME) in vivo remains insufficient. To facilitate the clinical translation of MGFD, further research is warranted to conduct comprehensive pharmacokinetic/pharmacodynamic studies alongside in vivo antitumor efficacy evaluations to thoroughly assess its safety and therapeutic potential.

## 4. Materials and Methods

### 4.1. Preparation of MGFD and UPLC-MS/MS Analysis

The MGFD consists of Buguzhi, Gusuibu, Guizhi, Fuling, Chishao, Mudanpi, and Taoren at a weight ratio of 1:1:1:1:1:1:1 (Table 2). All the crude herbs were from Beijing Tong Ren Tang (Chengdu, China). The herbs were combined and macerated in 75% ethanol (Chron Chemicals, Chengdu, China). The combination was subsequently exposed to reflux extraction using 75% ethanol, a process that was carried out three times. Following this, the resulting extracts underwent concentration via evaporation. The concentrated solution was then dried in a vacuum, ground into a fine powder, and preserved at −80 °C. At the point of use, the powder was sonicated in dimethyl sulfoxide (DMSO, Solarbio, Beijing, China).

High-resolution qualitative analysis of the constituents in MGFD was performed using the hybrid quadrupole-Orbitrap high-resolution mass spectrometer (UPLC-Q-Orbitrap HRMS) system (Thermo Fisher Scientific, Waltham, MA, USA). Prior to analysis, the UPLC-Q-Orbitrap HRMS system underwent strict calibration to ensure accuracy. Mass error in the generated mass spectrometry data was expressed in parts per million (ppm). The identified compounds exhibited an absolute mass error of less than 5 ppm, indicating high reliability of the identification results. Subsequently, the mass spectra were matched against the mzCloud online database and the in-house traditional Chinese medicine component database (OTCML) for confirmation. This process enabled the identification of key components in MGFD and validated the authenticity and reliability of the compound herbal materials. Total ion current is shown in Appendix A. Appendix A summarizes the main components identified in the seven herbs comprising the MGFD. Chromatographic conditions are provided in the Appendix A.

### 4.2. Cell Culture

Human PC3 and DU145 cells were from CELLCOOK (Guangzhou, China) and BOSTER (Wuhan, China), respectively. PC3 cells, linked to bone metastasis in PCa, were grown in F12K medium (BasalMedia, Shanghai, China) enriched with 10% fetal bovine serum (FBS, GeminiBio, West Sacramento, CA, USA), along with 100 U/mL of penicillin and 100 μg/mL of streptomycin (HyClone™, Logan, UT, USA). Meanwhile, DU145 cells, which are associated with brain metastasis, were maintained in RPMI 1640 medium (Gibco, Grand Island, CA, USA) that also contained 10% FBS (GeminiBio, West Sacramento, CA, USA), as well as 100 U/mL penicillin and 100 μg/mL streptomycin (HyClone™, Logan, UT, USA). The cells were kept under conditions of 37 °C and 5% CO_2_. The culture medium was refreshed every two days, and the cells underwent passaging every three days.

### 4.3. Cell Counting Kit-8 Assay

The Cell Counting Kit-8 (CCK8, MedChemExpress, Shanghai, China) was utilized to evaluate the viability of cells. PC3 and DU145 cell lines were uniformly distributed in 96-well plates (Corning, NY, USA), seeded at 1 × 10^4^ cells per well, and incubated overnight. On the following day, various concentrations of MGFD, from 1000 μg/mL down to 5 μg/mL, were prepared and applied in sets of five wells. In addition, DMSO control and blank control groups were used. Following a 48-h treatment period, we introduced the CCK-8 reagent, dispensing 10 μL into every well. The cells that had been plated were then incubated for about 2 h. Ultimately, the relative fluorescence intensity for each well was assessed with a Varioskan^®^ Flash (Thermo Fisher Scientific, Waltham, MA, USA) at 450 nm. Additionally, the growth proportion was determined using the following method: Cell Viability (%) = [A (Compound +) − A (Blank)]/[A (DMSO +) − A (Blank)] × 100%.

### 4.4. Wound Healing Assay

PC3 and DU145 cell lines were placed in a 96-well plate (1 × 10^4^ cells for each well). After overnight incubation, a wound was generated using an automated scratch tool, and any cell debris was thoroughly rinsed off with PBS. Following that, the cells were grown in a medium devoid of FBS, enriched with different amounts of MGFD (75 and 50 μg/mL) for a duration of 24 h. The process of wound healing was monitored, and photographs were taken utilizing the Incucyte^®^ SX1 Live-Cell Analysis System (Sartorius, Göttingen, Germany).

### 4.5. Transwell Migration and Invasion Assay

PC3 and DU145 cells were seeded onto polycarbonate filter membranes (6.5 mm in diameter, 8 μm pore size) in the upper transwell chambers (5 × 10^4^ cells/chamber) (Corning, NY, USA) containing serum-free cell suspension. Prior to the invasion assays, a matrix gel (Corning, NY, USA) was added to the upper chamber. The lower chamber was filled with conditioned medium containing 10% FBS. After incubation periods of 24 or 48 h, the inserts were carefully removed with forceps, the residual liquid in the upper chamber was blotted dry, and the chamber was washed gently once with PBS before transferring to a well preloaded with 800 μL of 4% paraformaldehyde for fixation. Non-invading cells were gently wiped from the top side of the transwell membrane with a cotton swab. Those that had moved to the lower membrane surface were then fixed, treated with crystal violet for 15 min, rinsed twice with PBS, and allowed to air dry. Ultimately, the cells were counted under a microscope across eight distinct observation fields to determine the average values.

### 4.6. HTS^2^ Assay and Data Processing

HTS^2^ represents a method for high-throughput screening that allows for the creation of extensive cellular gene expression profiles when subjected to herbal influences. The HTS^2^ assays mainly consist of cell culture, the design of probes, and the screening process. This research utilized HTS^2^ assays to investigate how seven different herbal extracts affect the expression of 3407 genes in both PC3 and DU145 cell lines.

In the process of sample manipulation, PC3 and DU145 cells were cultured in 384-well plates (2 × 10^3^ cells per well) for 24 h. Following this, the cells received treatment with seven different herbal extracts at varying concentrations of 100, 75, 50, and 25 μg/mL for an additional 24 h. The samples underwent automatic analysis utilizing the HTS^2^ platform and were sequenced using an Illumina HiSeq X Ten sequencer (Illumina, San Diego, CA, USA).

In the course of data processing, all reads were aligned to probe sequences and adjusted in relation to the expression levels of 18 stable housekeeping genes [52]. Following this, Pearson correlation coefficients were computed for the normalized transcriptional data utilizing R software, after conducting treatments with 11 DMSO replicates, 12 JQ1 replicates, and 4 replicates of the seven herbs in the MGFD. This analysis aimed to evaluate the reliability and consistency of the gene expression profiles. JQ1 was employed as an internal quality control of the HTS^2^ assay. Correlation coefficients that surpass 0.9 validate the consistency and reproducibility of the HTS^2^ assay.

Differential expression analysis was conducted using the DESeq2 R package (version 1.42.1) [53]. Initially, lowly expressed genes were filtered out from the raw count matrix. Sample groups were defined, and a DESeqDataSet object was constructed accordingly. The DESeq function was then applied to automatically perform normalization based on size factors, estimate dispersion, and fit a negative binomial generalized linear model. Differential expression was assessed using the Wald test. Genes exhibiting |FC| > 1.5 and *p*-values < 0.05 were identified as DEGs. In conclusion, heatmaps and volcano plots for these DEGs were generated with the help of the R packages pheatmap and ggplot2 [54], respectively.

### 4.7. GO and KEGG Enrichment Analysis

The Kyoto Encyclopedia of Genes and Genomes (KEGG), along with Gene Ontology (GO) enrichment analyses, can uncover alterations in biological pathways and functions resulting from the DEGs. The R packages ClusterProfiler (version 3.18.1), DOSE (version 3.16.0), and enrichplot (version 1.10.2) were applied. To visualize the KEGG networks, CytoScape version 3.9.0 was used, and the topological analysis of these networks was conducted with the CytoNCA package.

### 4.8. Protein-Protein Interaction Analysis

The STRING database (https://cn.string-db.org/) [55] was employed to obtain the network of protein-protein interactions (PPI) for the targets. Following this, PPI networks were created with a cumulative score greater than 0.9. Visualization of these networks was facilitated through Cytoscape software version 3.9.0, while the topological analysis was performed utilizing the CytoNCA software package. The initial evaluation involved establishing the degree centrality (DC) values to exceed twice the median of all nodes. The closeness centrality and betweenness centralities were employed as secondary metrics for the analysis. The top 20 proteins were identified as key targets.

### 4.9. Gene Set Enrichment Analysis

Gene Set Enrichment Analysis (GSEA) was employed to assess how each herb influences different gene sets. This approach utilized a list of DEGs in each herb and gene set. The genes were matched to the ranked list, and subsequently, the running sum was calculated [56]. The normalized enrichment score was subsequently derived by normalizing the enrichment score, with a false discovery rate below 0.05 deemed indicative of statistical significance. The outcomes from GSEA analysis were visualized by the networkD3 (version 0.4.1) package in R, and represented through a Sankey diagram.

### 4.10. Acquisition of Active Components and Targets of MGFD

The chemical constituents of the MGFD were obtained from three distinct databases: Traditional Chinese Medicine System Pharmacology Database (https://old.tcmsp-e.com/tcmsp.php, accessed on 19 August 2025) [57], Traditional Chinese Medicine Integrated Database (https://ngdc.cncb.ac.cn/databasecommons/database/id/437, accessed on 19 August 2025) [58], and a chemistry database developed by the Shanghai Institute of Organic Chemistry, part of the Chinese Academy of Sciences (http://www.organchem.csdb.cn/scdb/default.asp, accessed on 19 August 2025) [59]. The active ingredients were evaluated according to the criteria of oral bioavailability (OB) being at least 30% and drug-likeness (DL) of 0.18 or higher. The 2D structures of these substances were from the PubChem database (https://pubchem.ncbi.nlm.nih.gov/, accessed on 19 August 2025) [60]. Ultimately, the potential targets linked to these compounds were determined through Swiss Target Prediction (http://swisstargetprediction.ch/, accessed on 19 August 2025) [61].

### 4.11. Acquisition of PCa Targets and Construction of the Network

By employing the search terms "prostate cancer" and "metastatic castration-resistant prostate cancer," we conducted a search through the OMIM (https://omim.org/), GeneCards (http://www.genecards.org/) [62] and DisGeNET (https://disgenetplus.com/) [63] databases to uncover possible therapeutic targets for PCa. Overlapping portions from three databases were used as the set of disease targets. Following that, the Venn Online tool (http://bioinformatics.psb.ugent.be/, accessed on 19 August 2025) was utilized to depict the overlap between disease targets and drug targets. Finally, a visualization network of herb-active component-treatment targets was constructed using Cytoscape 3.9.0 [64].

### 4.12. Genetic Alteration Analysis

cBioPortal (accessible at https://www.cbioportal.org/, with the latest update on August 21, 2023) is an online integrated data mining platform that was employed to investigate genetic modifications and conduct survival analysis concerning MGFD-related targets in the context of prostate cancer treatment [65]. This study included six prostate cancer cohorts from the cBioPortal database, primarily derived from multiple research projects such as MSK, MCTP, Broad/Cornell and the SU2C/PCF Dream Team [19,20,21,22,23,24]. The total sample size comprised 3679 patients, encompassing 2964 non-metastatic and 715 metastatic cases, thereby covering various disease stages.

### 4.13. Disease-Free Survival Analysis

The Gene Expression Profiling Interactive Analysis (GEPIA2) [66] tool was utilized to conduct an analysis of disease-free survival concerning the primary targets linked to MGFD in prostate cancer. This comprehensive examination seeks to elucidate the connection between gene expression levels and patient outcomes, providing important insights into the elements influencing disease-free survival in this type of cancer.

### 4.14. Molecular Docking for Targets Verification

In this research, the crystal structures of essential target proteins were from PDB (https://www.rcsb.org/) [67]. The preparation of proteins was carried out utilizing the "Protein Preparation Wizard" feature of the Schrödinger software (Schrödinger, 2018). Subsequently, the LigPrep module was employed to create three-dimensional structures and minimize the energy needed for the functional components of the MGFD. The receptor mesh generation module was used to form the mesh box, and Glide XP docking software was used to calculate the docking results. Finally, 3D visualization of protein-ligand interactions was performed using PyMOL 2.6.0.

### 4.15. Statistical Analysis

A statistical examination of the experimental data was conducted three times utilizing Student's *t*-test. The results are shown as the average along with the standard error. Quantitative findings were illustrated with the GraphPad Prism software (version 9.0). Significance levels were set at * *p* < 0.05, ** *p* < 0.01, and *** *p* < 0.001.

### 4.16. Quantitative Real-Time Polymerase Chain Reaction (qRT-PCR) Assay

Total RNA was isolated from cells using the Cell Total RNA Isolation Kit (Vazyme, Nanjing, China), and subsequently reverse transcribed into cDNA with the cDNA Synthesis Kit (Vazyme, Nanjing, China). Quantitative real-time PCR (qRT-PCR) was performed on a Bio-Rad CFX Duet Fluorescent Quantitative PCR System using the ChamQ SYBR qPCR Master Mix (Vazyme, Nanjing, China). The thermal cycling protocol consisted of an initial denaturation at 95 °C for 15 min, followed by 40 cycles of 95 °C for 10 s, 59 °C for 20 s, and 72 °C for 30 s. Relative expression levels of target genes were determined using the 2^−ΔΔCt^ method, with β-actin mRNA serving as the internal control for normalization.

## 5. Conclusions

Our research revealed that MGFD effectively hinders the proliferation, migration, and invasion of human PCa cells mainly by influencing the FoxO signaling pathway, promoting cellular senescence, and affecting the TNF signaling pathway. Key targets of MGFD in the treatment of PCa include AKT1, CASP8, CDK1, and CCND1. The primary potential bioactive compounds identified through molecular docking that contribute to the inhibition of PCa cells are baicalein, quercetin, and 5-[[5-(4-methoxyphenyl)-2-furyl] methylene] barbituric acid. The results emphasize MGFD's potential as an effective adjuvant in treating PCa (Figure 11). Furthermore, extensive transcriptional analysis uncovered the main function of each herb within MGFD. To summarize, this research uncovers the mechanisms of MGFD that act through multiple target and pathways in PCa treatment, while also identifying compounds that may possess anticancer properties. The profile of its multi-target activity highlights its potential as a therapeutic option for PCa, encouraging additional validation studies in vivo.

## Figures and Tables

**Figure 1 pharmaceuticals-18-01275-f001:**
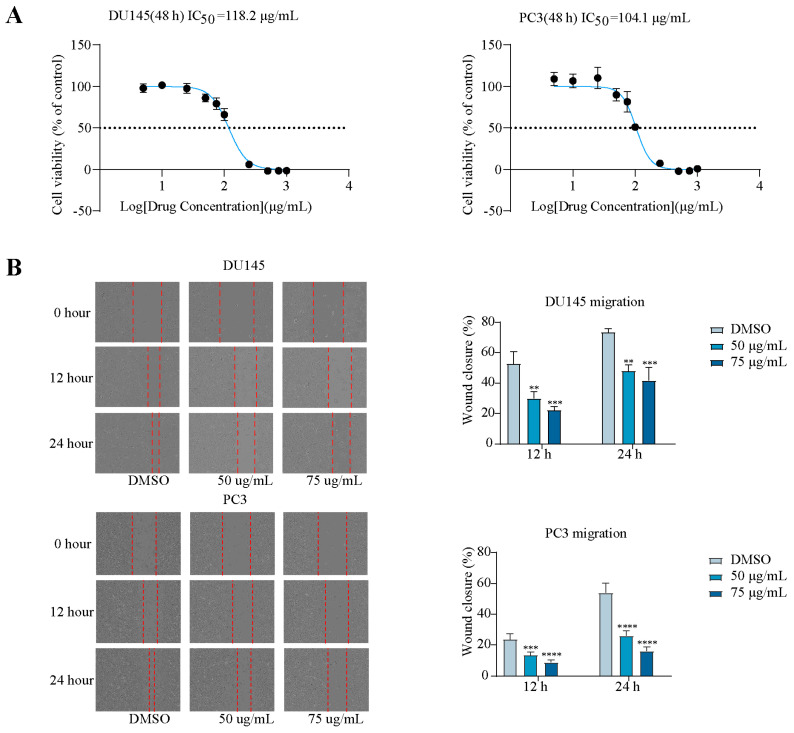
MGFD inhibits malignant phenotype in PCa cells. (**A**) Proliferation ability was detected through CCK-8 in DU145 and PC3 cells treated with MGFD. (**B**) Wound healing assay evaluated the inhibitory effect of MGFD at 75 and 50 μg/mL in DU145 and PC3 cells. (**C**) Transwell assays demonstrate the effects of MGFD at 75 and 50 μg/mL on migration and invasion in DU145 and PC3 cells. DMSO, dimethyl sulfoxide. Data are presented as mean ± SD, **** *p* < 0.0001, *** *p* < 0.001, ** *p* < 0.01.

**Figure 2 pharmaceuticals-18-01275-f002:**
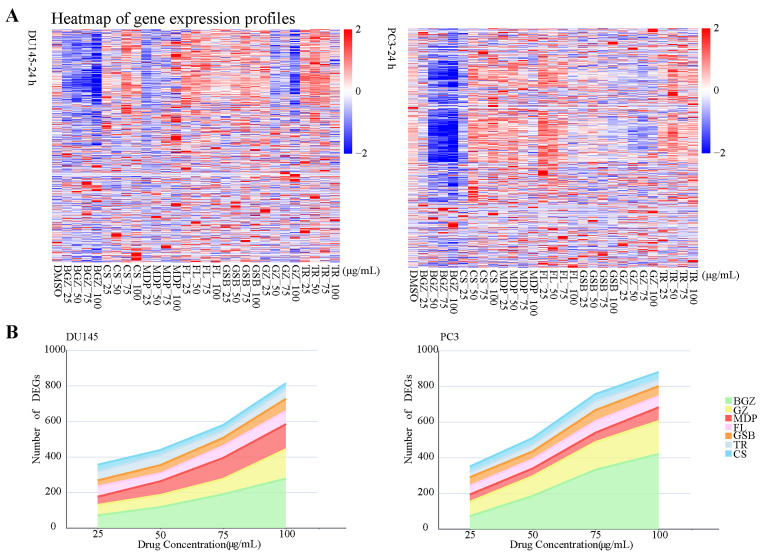
HTS^2^ assay results. (**A**), Heatmap of gene expression profiles for four drug concentrations (100, 75, 50, 25 μg/mL) of seven herbs and a control group (DMSO group) obtained through HTS^2^ assay treatment in DU145 and PC3 cells. (**B**), Line graph illustrating the variation in DEGs following treatment with the herbs at four concentrations in DU145 and PC3 cells. (FL, Fuling; MDP, Mudanpi; BGZ, Buguzhi; GZ, Guizhi; GSB, Gusuibu; CS, Chishao; TR, Taoren). The criteria for DEGs were set at |Foldchange| ≥ 1.5 and *p*-value < 0.05.

**Figure 3 pharmaceuticals-18-01275-f003:**
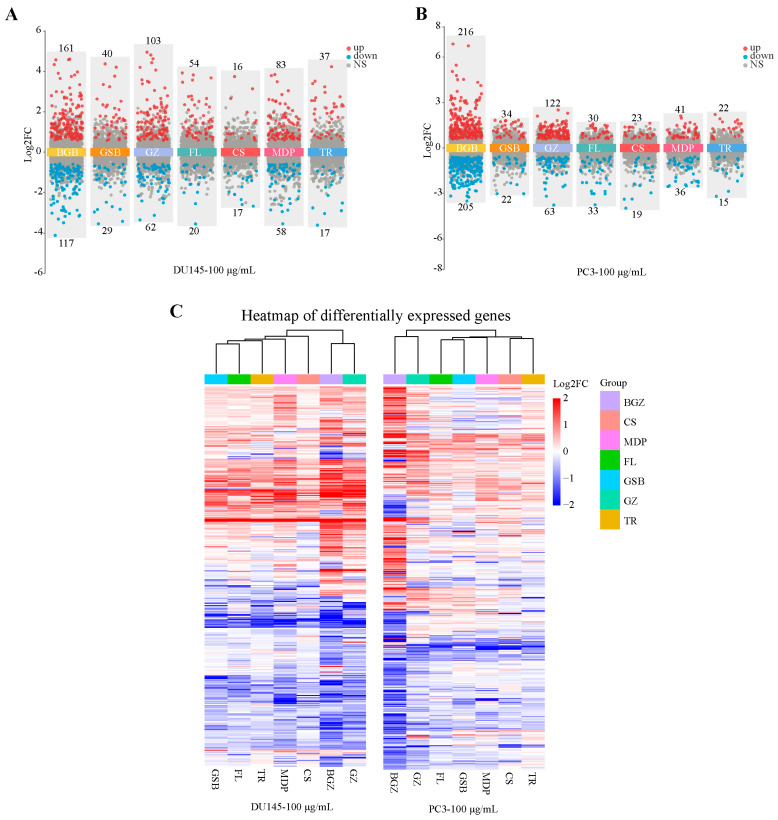
DEGs analysis. Volcano plot of DEGs in (**A**) DU145 and (**B**) PC3 cells treated with 100 μg/mL. The criteria for DEGs were set at |Foldchange| ≥ 1.5 and *p*-value < 0.05. Red and green dots indicate upregulated and downregulated DEGs, respectively. (**C**), Heatmap of DEGs in the DU145 and PC3 cells treated with 100 μg/mL drug, clustering the seven herbs based on DEG similarities and differences. (FL, Fuling; MDP, Mudanpi; BGZ, Buguzhi; GZ, Guizhi; GSB, Gusuibu; CS, Chishao; TR, Taoren).

**Figure 4 pharmaceuticals-18-01275-f004:**
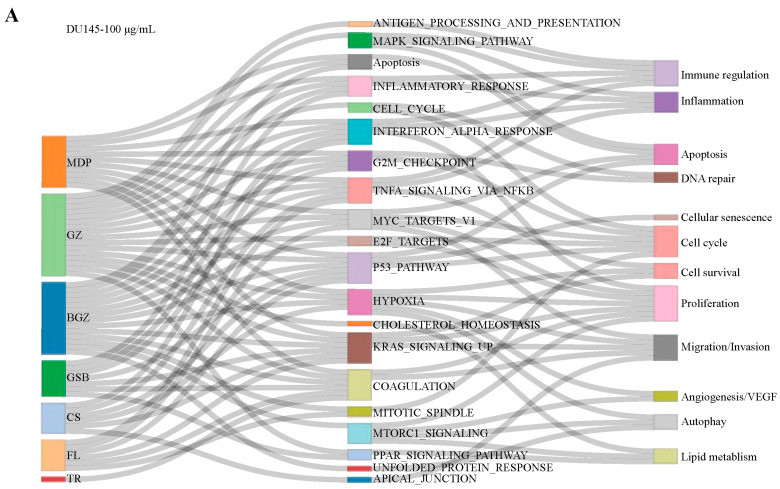
GSEA enrichment analysis. (**A**,**B**), The Sankey diagram associated with herbs, gene sets, and biological function in DU145 and PC3 cells. Grey lines represent the relationships among herbs, gene sets, and biological functions. (**C**,**D**), Lollipop chart illustrated shared pathways in DU145 and PC3 cells. (FL, Fuling; MDP, Mudanpi; BGZ, Buguzhi; GZ, Guizhi; GSB, Gusuibu; CS, Chishao; TR, Taoren).

**Figure 5 pharmaceuticals-18-01275-f005:**
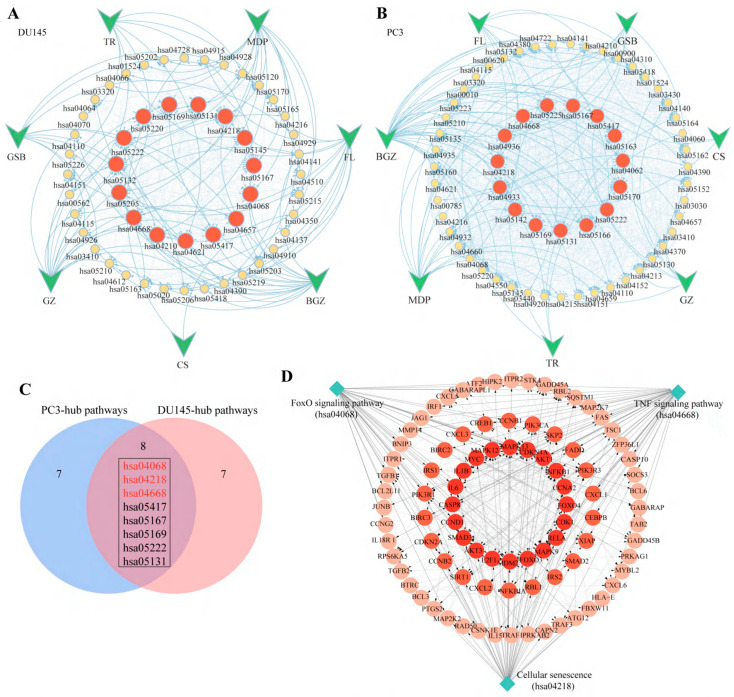
Herb-KEGG pathway interaction network and GO enrichment analysis. (**A**,**B**), KEGG interaction network of the seven herbs in DU145 and PC3 cells, respectively. Grey lines represent the interactions among pathways. (FL, Fuling; MDP, Mudanpi; BGZ, Buguzhi; GZ, Guizhi; GSB, Gusuibu; CS, Chishao; TR, Taoren). (**C**), Venn diagram of hub pathways shared by DU145 and PC3 cells. (**D**), Interaction network analysis of hub pathways and targets. Red represents the top 20 hub targets with the highest degree scores. Grey lines represent the relationships among pathways and genes. (**E**,**F**), GO enrichment analysis of DEGs in MGFD in DU145 and PC3 cells. BP, Biological Process. CC, Cellular Component. MF, Molecular Function.

**Figure 6 pharmaceuticals-18-01275-f006:**
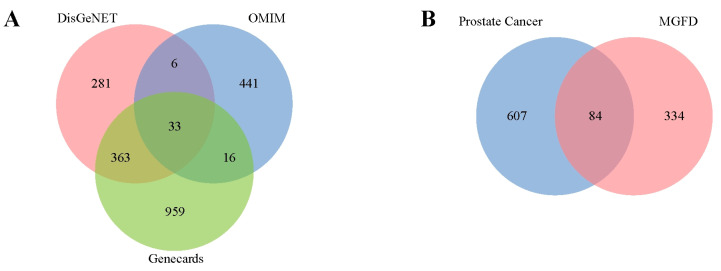
Venny map. (**A**), Venn diagram of potential disease targets for PCa. (**B**), Overlap of MGFD drug targets and PCa disease targets.

**Figure 7 pharmaceuticals-18-01275-f007:**
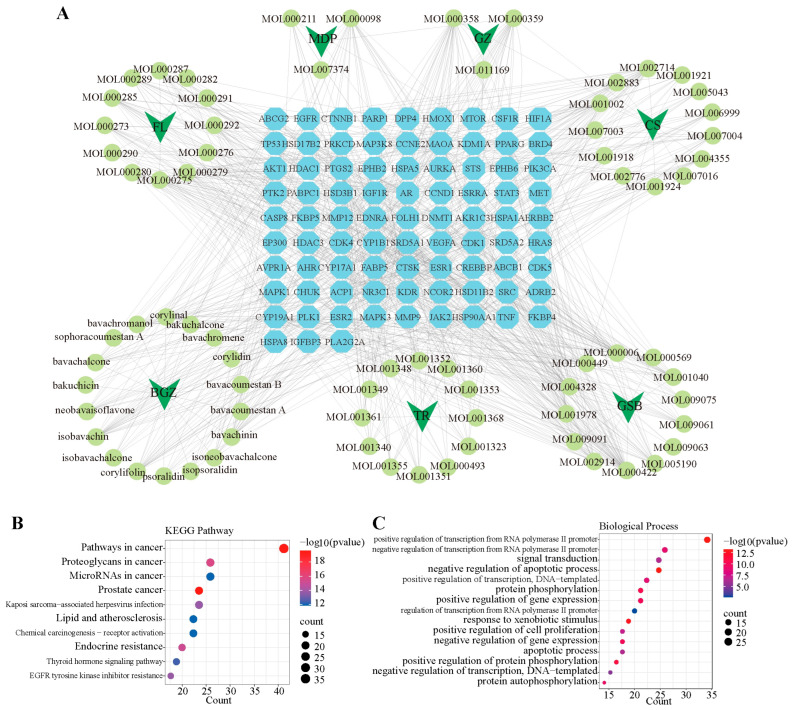
Network pharmacology analysis of MGFD. (**A**), Active component-target networks of MGFD, where blue and green nodes indicate potential therapeutic targets and active components of MGFD (FL, Fuling; MDP, Mudanpi; BGZ, Buguzhi; GZ, Guizhi; GSB, Gusuibu; CS, Chishao; TR, Taoren), respectively. Grey lines represent the relationships among herbs, potential therapeutic targets, and active components. (**B**) KEGG pathway analysis and (**C**) GO analysis of 84 potential targets in MGFD.

**Figure 8 pharmaceuticals-18-01275-f008:**
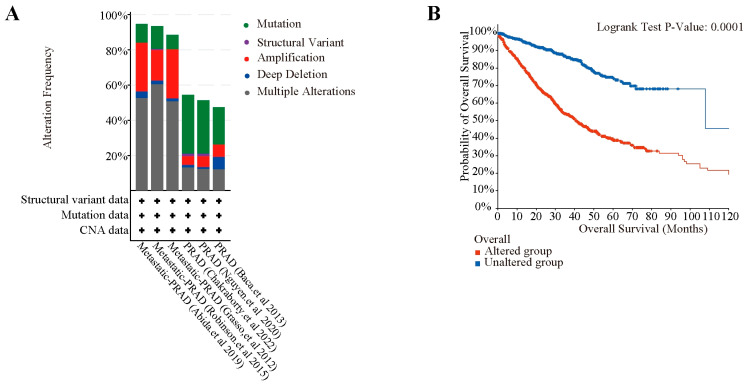
Genetic alteration and survival analysis of 84 therapeutic targets. (**A**), Genomic Alterations in the 84 potential therapeutic targets in six different PCa studies [19,20,21,22,23,24]. (**B**), Kaplan–Meier survival analysis comparing mutant and non-mutant groups among the 84 targets, with *p*-value < 0.05 indicating statistical significance.

**Figure 9 pharmaceuticals-18-01275-f009:**
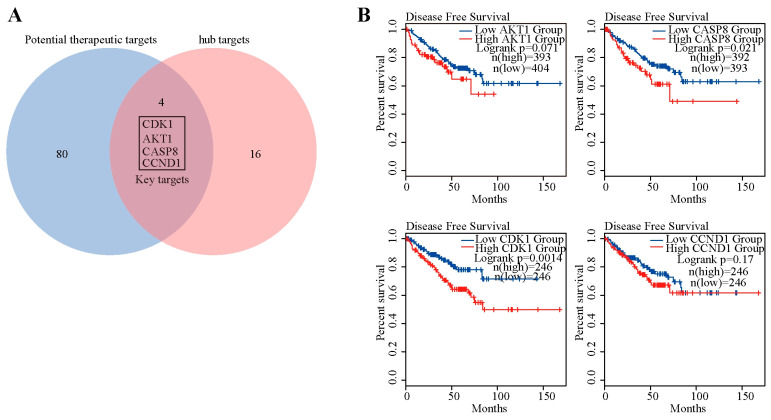
Disease-free survival analysis of four key targets. (**A**), The Venn diagram construction of the 84 therapeutic targets with the top 20 hub targets in the gene expression profiles. (**B**), Disease-free survival analyses on AKT1, CDK1, CCND1, and CASP8 in PCa.

**Figure 10 pharmaceuticals-18-01275-f010:**
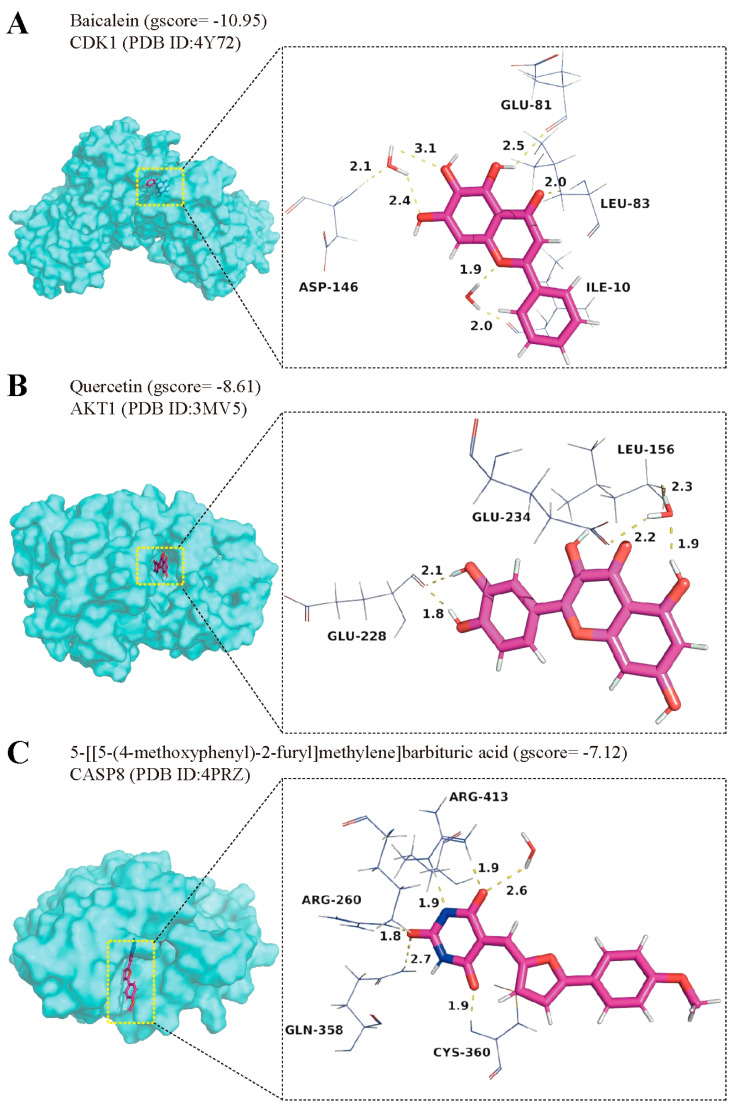
Three-dimensional (3D) interaction diagrams of molecular docking. (**A**) CDK1 with baicalein, (**B**) AKT1 with quercetin, (**C**) CASP8 with 5-[[5-(4-methoxyphenyl)-2-furyl] methylene] barbituric acid. Yellow dashed lines indicate hydrogen bonds. Purple sticks represent carbon atoms, red sticks represent oxygen atoms, and blue sticks represent hydrogen atoms.

**Figure 11 pharmaceuticals-18-01275-f011:**
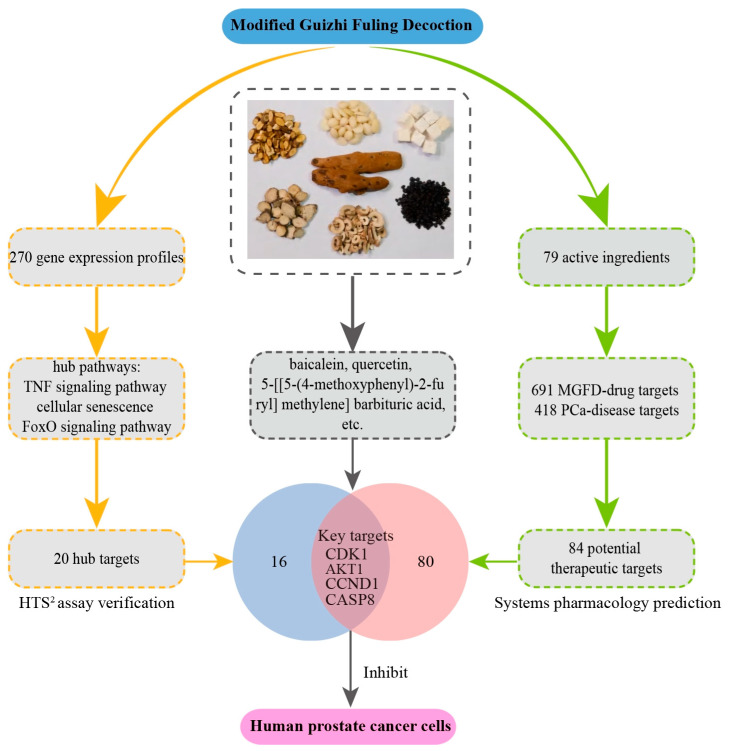
Overview of molecular mechanisms of MGFD in human PCa cells.

**Table 1 pharmaceuticals-18-01275-t001:** The results of molecular docking.

PubChem CID	Chemical Name	Glide Gscore
*AKT1(PDB ID: 3MV5)*		
5280343	Quercetin	−8.61
5281855	Ellagic acid	−8.36
440735	Eriodictyol	−6.76
6476086	Bakuchalcone	−6.63
5280445	Luteolin	−5.94
5468522	Bakuchiol	−5.75
5280863	Kaempferol	−5.68
373261	Eriodyctiol (flavanone)	−5.17
5318608	Isoneobavachalcone	−5.09
10181133	Cerevisterol	−4.26
*CASP8(PDB ID: 4PRZ)*		
2177166	5-[[5-(4-methoxyphenyl)-2-furyl] methylene] barbituric acid	−7.12
73981645	Lactiflorin	−6.49
*CDK1(PDB ID: 4Y72)*		
5281605	Baicalein	−10.95
5280343	Quercetin	−10.82
5280445	Luteolin	−10.81
5281220	Aureusidin	−10.06
5280863	Kaempferol	−7.87
5320053	Neobavaisoflavone	−7.34
5321790	Bavachromanol	−5.05
14236566	Corylifolin	−4.72
10337211	Bavachinin	−4.65

Protein Data Bank (PDB).

**Table 2 pharmaceuticals-18-01275-t002:** Detailed information of herbs in MGFD.

Accepted Scientific Name	Chinese Name	Part(s) Used	Amount (g)
*Cullen corylifolium* (L.) Medik.	Buguzhi	Dried mature fruit	10
*Drynaria roosii* Nakaike	Gusuibu	Dried root	10
*Neolitsea cassia* (L.) Kosterm.	Guizhi	Dried branch	10
*Poria Cocos* (Schw.) Wolf.	Fuling	Dried sclerotium	10
*Paeonia lactiflora* Pall.	Chishao	Dried root	10
*Prunus persica* (L.) Batsch	Taoren	Dried seed	10
*Paeonia × suffruticosa* Andrews	Mudanpi	Dried root	10

## Data Availability

Data is contained within the article and Appendix A.

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
