# Peer review of "Large-Scale Transcriptome Profiling and Network Pharmacology Analysis Reveal the Multi-Target Inhibitory Mechanism of Modified Guizhi Fuling Decoction in Prostate Cancer Cells"

_pharmaceuticals, 2025, doi:10.3390/ph18091275_

Round 1

Reviewer 1 Report

Comments and Suggestions for Authors

Review Report

The present study aimed to investigate the mechanism of action (MOA) of a modified Traditional Chinese Medicine (TCM) herbal formulation (MGFD), which has been widely applied in clinical oncology. The authors employed high-throughput screening (HTS), network pharmacology approaches, and conventional in vitro cellular assays to investigate the MOA of MGFD. While the conclusions were drawn based on their findings, several critical issues concerning experimental design and data interpretation need to be addressed and discussed in greater detail.

Major Concerns

1. Concentration Range and Cytotoxicity in Functional Assays

The reported IC50 values of MGFD were 118.2 µg/mL for DU145 cells and 104.1 µg/mL for PC3 cells. However, the authors conducted cell migration and invasion assays using concentrations of 50, 75, and 100 µg/mL of MGFD. They reported dose-dependent inhibitory effects in both assays at these concentration ranges. Given the proximity of these concentrations—particularly 100 µg/mL—to the IC50 values, it is likely that significant cytotoxicity was present, potentially confounding the interpretation of migration and invasion inhibition studies. Such cellular functional assays should be performed within a non- or minimal cytotoxic concentration range, typically where cell viability remains above 80–90%. This limitation should also be discussed in the context of the HTS2 and network pharmacology studies.

2. Lack of Experimental Validation of Predicted Targets

The 3 key active components and their predicted molecular targets, identified via network pharmacology, should be experimentally validated using appropriate in vitro assays. Suggested validation methods include biochemical activity assays, immunoblotting for target or pathway modulation, and cell proliferation or viability assays. Without such experimental confirmation, the conclusions regarding the mechanism of action remain speculative and should be interpreted with caution.

Minor Comments

Lines 96–98: This section is better suited for the Discussion and should be relocated accordingly or deleted.

Supplementary Tables and Figure: These should be sequentially numbered according to their order of appearance in the main text.

Figure 2: The abbreviations for the herbal components in the figure legend should be consistent with those used in the Figure 2 and throughout the manuscript.

Lines 157–158: A more detailed explanation of the underlying TCM theory is recommended for clarity and completeness.

Lines 183–184: The statement referencing specific findings is unclear. Please indicate where this result can be found within the manuscript.

Lines 279–280: The listed chemicals and their corresponding targets should be corrected to match the information presented in Table 1 and Figure 10.

Reviewer 2 Report

Comments and Suggestions for Authors

This manuscript aims to elucidate the mechanistic basis of Modified Guizhi Fuling Decoction (MGZFLD) in prostate cancer (PCa) through RNA-Seq-based transcriptomics and network pharmacology analysis. Some noteworthy critics are given below:

-The RNA-Seq data reveal differentially expressed genes (DEGs), but no qPCR or protein-level validation (e.g., Western blot) is provided.

-The plant author names should not be italicized.

-Include the plant parts used for each plant species in the formulation.

-Is this formulation registered in the Chinese Pharmacopeia? Please indicate.

-Although MGZFLD is applied to PCa cells, the paper does not report cell proliferation, apoptosis, migration, or viability assays.

-Volcano plots and PCA figures are used, but adjusted p-values should be clearly specified.

-While multiple targets and pathways are mentioned, these are predicted in silico without empirical confirmation.

-Language quality is moderate; consider professional English editing.

-Provide full names of abbreviations at first use (e.g., MGZFLD, TCMSP, DEG).

-Network interpretation used in the paper seems based on over-reliance on predicted targets without empirical support.

Reviewer 3 Report

Comments and Suggestions for Authors

The comments are attached

Round 2

Reviewer 1 Report

Comments and Suggestions for Authors

Majors

The revised manuscript still fails to justify the use of the maximum concentration of MGFD and its 7 individual plant extracts for transcriptomic analysis. The authors used 100 µg/mL MGFD as a maximunm concentration on both DU15 and PC12 cells, resulting in 66% and 51% viability, respectively, indicating significant cytotoxicity. In addition, the cytotoxicity data for 7 plant extracts were not presented in the manuscript. Excessive toxicity—such as cell deaths or mRNA translation inhibition—can obscure the determine the intrinsic MoA) of the drug by amplifying general stress or metabolic suppression signatures rather than drug-specific signals. Therefore, it is recommended that the highest concentration be non-cytotoxic or within a tolerable range. (Ref. 10.1021/tx400402j).

Minors

-JQ1 (Line 481) should be fully describe and explain the rationale for using this compound in the transcriptomic analysis.

Reviewer 2 Report

Comments and Suggestions for Authors

The revision request, which was at a mild level, was performed well. In my opinion, it might be accepted now.

Comments on the Quality of English Language

Can be improved.

Reviewer 3 Report

Comments and Suggestions for Authors

The author has  responded to all comments provided

Round 3

Reviewer 1 Report

Comments and Suggestions for Authors

The authors have addressed all major concerns. The final revised manuscript is now acceptable for publication.